# Atf3 defines a population of pulmonary endothelial cells essential for lung regeneration

**Terren K Niethamer**[1,2,3,4], **Lillian I Levin**[1,4], **Michael P Morley**[1,3,4], **Apoorva Babu**[1,3,4], **Su Zhou**[1,4], **Edward E Morrisey**[1,2,3,4]*

[1]Department of Medicine, Philadelphia, United States; [2]Department of Cell and Developmental Biology, Philadelphia, United States; [3]Penn-Children's Hospital of Philadelphia Lung Biology Institute, University of Pennsylvania, Philadelphia, United States; [4]Penn Cardiovascular Institute, University of Pennsylvania, Philadelphia, United States

**Abstract** Following acute injury, the capillary vascular bed in the lung must be repaired to reestablish gas exchange with the external environment. Little is known about the transcriptional and signaling factors that drive pulmonary endothelial cell (EC) proliferation and subsequent regeneration of pulmonary capillaries, as well as their response to stress. Here, we show that the transcription factor Atf3 is essential for the regenerative response of the mouse pulmonary endothelium after influenza infection. *Atf3* expression defines a subpopulation of capillary ECs enriched in genes involved in endothelial development, differentiation, and migration. During lung alveolar regeneration, this EC population expands and increases the expression of genes involved in angiogenesis, blood vessel development, and cellular response to stress. Importantly, endothelial cell-specific loss of Atf3 results in defective alveolar regeneration, in part through increased apoptosis and decreased proliferation in the endothelium. This leads to the general loss of alveolar endothelium and persistent morphological changes to the alveolar niche, including an emphysema-like phenotype with enlarged alveolar airspaces lined with regions that lack vascular investment. Taken together, these data implicate Atf3 as an essential component of the vascular response to acute lung injury that is required for successful lung alveolar regeneration.

*For correspondence:
emorrise@pennmedicine.upenn.edu

## Editor's evaluation

The reviewers concurred that your study has advanced our mechanistic understanding of lung regeneration. While the importance of regeneration of alveolar capillaries for long response to injury has been long recognized, the regulation of this process has not been well understood. You provide novel, comprehensive, and compelling evidence that the expression of the transcription factor Atf3 in alveolar capillary endothelial cells plays a critical role in the regeneration of alveolar capillaries following lung injury.

## Introduction

The cardinal function of the mammalian lung is gas exchange with the cardiovascular system, which requires the close interface of pulmonary capillary ECs and alveolar epithelial type 1 (AT1) cells (*Zepp and Morrisey, 2019*). The functional unit of gas exchange in the distal lung is the alveolus, where large, flat AT1 cells stretch across a network of capillaries to mediate this process. Although the capillary-AT1 interface is essential for respiration and has been beautifully imaged in three dimensions at

the macroscopic level (*Weibel, 2017*), how it is formed and maintained at a molecular level remains poorly understood. Upon acute injury, the gas exchange interface must be rapidly repaired to restore lung function, a process that requires regeneration of cell types lost to injury, orchestration of proper communication between them to initiate an exchange of oxygen and carbon dioxide and reestablishment of the functional architecture. Given the central role that capillary endothelium plays in lung function, it represents an attractive target for regenerative therapies. However, a deeper understanding of the mechanisms of EC regeneration is needed to develop such targeted therapies.

Recent studies have demonstrated that the lung endothelium is heterogeneous and that pulmonary capillary EC subtypes likely play different roles in progenitor function, signaling, and gas exchange (*Gillich et al., 2020*; *Jambusaria et al., 2020*; *Kalucka et al., 2020*; *Niethamer et al., 2020*; *Vila Ellis et al., 2020*). The CAP1 (*Plvap* cell or general capillary) EC population is the primary contributor to EC proliferation during the response to elastase injury (*Gillich et al., 2020*), while the CAP2 (*Car4* cell or aerocyte) is generated late in embryonic lung development (*Vila Ellis et al., 2020*) and is larger and more complex (*Gillich et al., 2020*; *Niethamer et al., 2020*; *Sun et al., 2022*; *Vila Ellis et al., 2020*). CAP2s also express genes that suggest they may possess a specialized function in gas exchange (*Gillich et al., 2020*). However, the molecular drivers that enable progenitor function in CAP1s remain elusive, and the factors that promote proliferation in these cells are unknown. Furthermore, if CAP1 proliferation is required to regenerate the entire capillary vasculature after acute lung injury, then a switch between CAP1 proliferation and CAP1-to-CAP2 differentiation may also be necessary. However, to date, no candidate mediators of proliferation or differentiation in CAP1s have been identified.

During lung development and regeneration, transcription factor (TF) expression within the alveolar epithelium regulates cell lineage commitment to maintain a balance between the proliferation of surviving cells and differentiation to replace cells that have been lost to injury (*Khattar et al., 2022*; *Liberti et al., 2022*; *Little et al., 2021*; *Penkala et al., 2021*; *Yang et al., 2015*; *Zhong et al., 2022*). Differential expression of TFs in the pulmonary vascular endothelium is also likely to play key roles in maintaining progenitor function, driving the proliferation of one pulmonary EC subtype while maintaining another in a differentiated state. Temporal changes in TF expression may also alter the choice to proliferate or differentiate within a certain cell population. Specific TFs also differentially regulate the activation of signaling pathways in the lung known to affect progenitor function, including Wnt signaling (*Barkauskas et al., 2013*; *Frank et al., 2016*; *Li et al., 2022*; *Liberti et al., 2021*; *Nabhan et al., 2018*; *Wang et al., 2016*; *Zacharias et al., 2018*).

We previously showed that the TF activating transcription factor 3, or *Atf3* is differentially expressed between CAP1 and CAP2 ECs (*Niethamer et al., 2020*) and could represent a key regulator of proliferation and CAP1-CAP2 differentiation. *Atf3* is an immediate early gene known to play essential roles in the maintenance of homeostasis, regeneration, and the response to cellular stress. Loss of *Atf3* expression is associated with increased tumorigenesis (*Wang et al., 2018*) and cancer metastasis (*Wolford et al., 2013*). *Atf3* expression can also be protective in situations of cellular stress. It is transcribed in response to DNA damage (*Abe et al., 2003*), after which it can inhibit apoptosis (*Hamdi et al., 2008*). Atf3 exhibits this protective role in the kidney (*Yoshida et al., 2008*), the intestine (*Zhou et al., 2017*), and the airway in asthma (*Gilchrist et al., 2008*). In addition to its role in stress response, Atf3 is also known to drive regeneration directly. In the nervous system, *Atf3* expression regulates axon sprouting and regeneration after nerve injury (reviewed in *Katz et al., 2022*). Mice lacking *Atf3* have reduced induction of regeneration-associated genes after injury to the peripheral nerves and are unable to appropriately regenerate their axons (*Gey et al., 2016*). More recently, Atf3 has been shown to drive regeneration of the infra-renal aorta after mechanical pressure injury (*McDonald et al., 2018*). *Atf3* forms part of a group of genes induced upon injury, including *Fos*, *Jun*, *Klf2*, and *Klf4*, and is required for endothelial proliferation to repair the aorta (*McDonald et al., 2018*). The role of Atf3 in tissue protection and induction of regeneration in other contexts, including the lung vasculature, remains unknown.

Here, we show that the expression of *Atf3* defines a subset of CAP1s that contribute to regeneration in the distal lung after acute influenza infection. Atf3-expressing CAP1s proliferate and increase in number after influenza infection, and endothelial-specific loss of Atf3 results in the persistent dysmorphic alveolar structure during the tissue repair process. Analysis of the transcriptional changes caused by loss of Atf3 reveals that it regulates alveolar regeneration by controlling the response both to the

stress of tissue injury and to regenerative cues. Loss of Atf3 results in increased endothelial apoptosis, decreased proliferation, and an overall persistent loss of ECs in Atf3-knockout animals. Loss of ECs and failure to regenerate the lung vasculature results in decreased density of alveolar epithelial cells and CAP2s, and a persistent emphysema-like phenotype characterized by enlarged alveolar airspaces. These data demonstrate that endothelial repair is critical to the structural integrity of regenerated lung tissue after influenza and that this process is regulated by Atf3.

## Results

### Expression of *Atf3* defines a subset of pulmonary capillary endothelial cells

CAP1 ECs in the distal lung alveolus proliferate after acute viral or elastase injury (*Gillich et al., 2020*; *Niethamer et al., 2020*), but the mechanisms that induce CAP1 proliferation, as well as the role of CAP1 proliferation in the regeneration of the overall lung tissue structure, remain incompletely understood. To determine factors that promote CAP1 response to acute lung injury, we analyzed single-cell RNA sequencing (scRNA-seq) data from non-immune (CD45-negative) cells of the whole mouse lung at homeostasis and 14 days post-H1N1 PR8 influenza injury (*Niethamer et al., 2020*). We subclustered endothelial populations expressing the pan-EC marker *Pecam1* (CD31) and observed endothelial populations previously described both at homeostasis and after injury, including lymphatic, arterial, and venous ECs as well as CAP1 and CAP2 ECs (*Gillich et al., 2020*; *Kalucka et al., 2020*; *Niethamer et al., 2020*; *Sun et al., 2022*; *Vila Ellis et al., 2020*; *Figure 1A–B*; *Figure 1—figure supplement 1A-G*; *Figure 1—figure supplement 2A-H*). Of note, subclustering of *Pecam1*+ cells within this dataset revealed two subpopulations of CAP1s (CAP1_A and CAP1_B) present both at homeostasis and after H1N1, with differing proportions of each CAP1 subcluster in the uninjured and injured lung (*Figure 1A and B*). CAP1_As demonstrated a more general gene expression profile shared by both CAP1_As and CAP1_Bs and were defined by expression of *Gpihbp1* and *Sema3c*, while CAP1_Bs possessed a more distinct gene expression profile defined by expression of *Atf3* and *Hes1* (*Figure 1A and B*; *Figure 1—figure supplement 1A, B, G, H*; *Figure 1—figure supplement 2A, B, H, I*). *Atf3* expression was highest in CAP1_Bs both at homeostasis and after injury (*Figure 1A′–A″ and B′–B″*; *Figure 1—figure supplement 1G*; *Figure 1—figure supplement 2H*). Proliferating ECs at 14 dpi also expressed *Atf3*, suggesting that expression of *Atf3* may be important for endothelial proliferation, which arises largely from the CAP1 population (*Gillich et al., 2020*; *Niethamer et al., 2020*; *Figure 1B′–B″*).

Integration of control and H1N1 scRNA-seq data also resulted in subclustering of CAP1_A and CAP1_B ECs (*Figure 1—figure supplement 3A.B*), which were similarly defined by expression of *Atf3* and *Hes1* (*Figure 1—figure supplement 3C*). Differential gene expression analysis comparing CAP1 EC populations at homeostasis and after H1N1 infection revealed that apoptotic signaling as well as regulation of angiogenesis, vascular development, and endothelial differentiation are upregulated in CAP1_As after H1N1, while regulation of angiogenesis, vascular development, endothelial cell migration, and leukocyte cell-cell adhesion are upregulated in CAP1_Bs after H1N1 (*Figure 1—figure supplement 3D, E*). At 14 days post-H1N1, *Atf3*-expressing CAP1_Bs made up a larger proportion of the CAP1 population, suggesting that the CAP1_B state may expand and contribute to the proliferative and regenerative responses of pulmonary ECs following acute lung injury (*Figure 1B*; *Figure 1—figure supplement 3B*).

To establish the role of *Atf3*-expressing CAP1_Bs and determine their function after lung injury, we performed differential gene expression analysis and gene ontology (GO) analysis comparing CAP1_A and CAP1_B subpopulations. Differential gene expression analysis at homeostasis revealed that CAP1_Bs express high levels of not only *Atf3*, but also the transcription factors *Hes1*, *Klf2*, and *Klf4*, as well as several heat shock proteins (*Figure 1—figure supplement 1H*). GO analysis suggested that the CAP1_B state at homeostasis is associated with the expression of genes involved in endothelial development and differentiation as well as cell migration, vascular development, and angiogenesis (*Figure 1C*). Differential gene expression analysis of CAP1_Bs at 14 days post-H1N1 shows that these cells express high levels of Atf3 target genes *Jun* and *Fos*, as well as the transcription factors *Hes1*, *Klf2*, and *Klf4*, several heat shock proteins, and the stress response genes *Dusp1* and *Ppp1r15a* (*Figure 1—figure supplement 2I*). GO analysis suggested that the CAP1_B regenerative

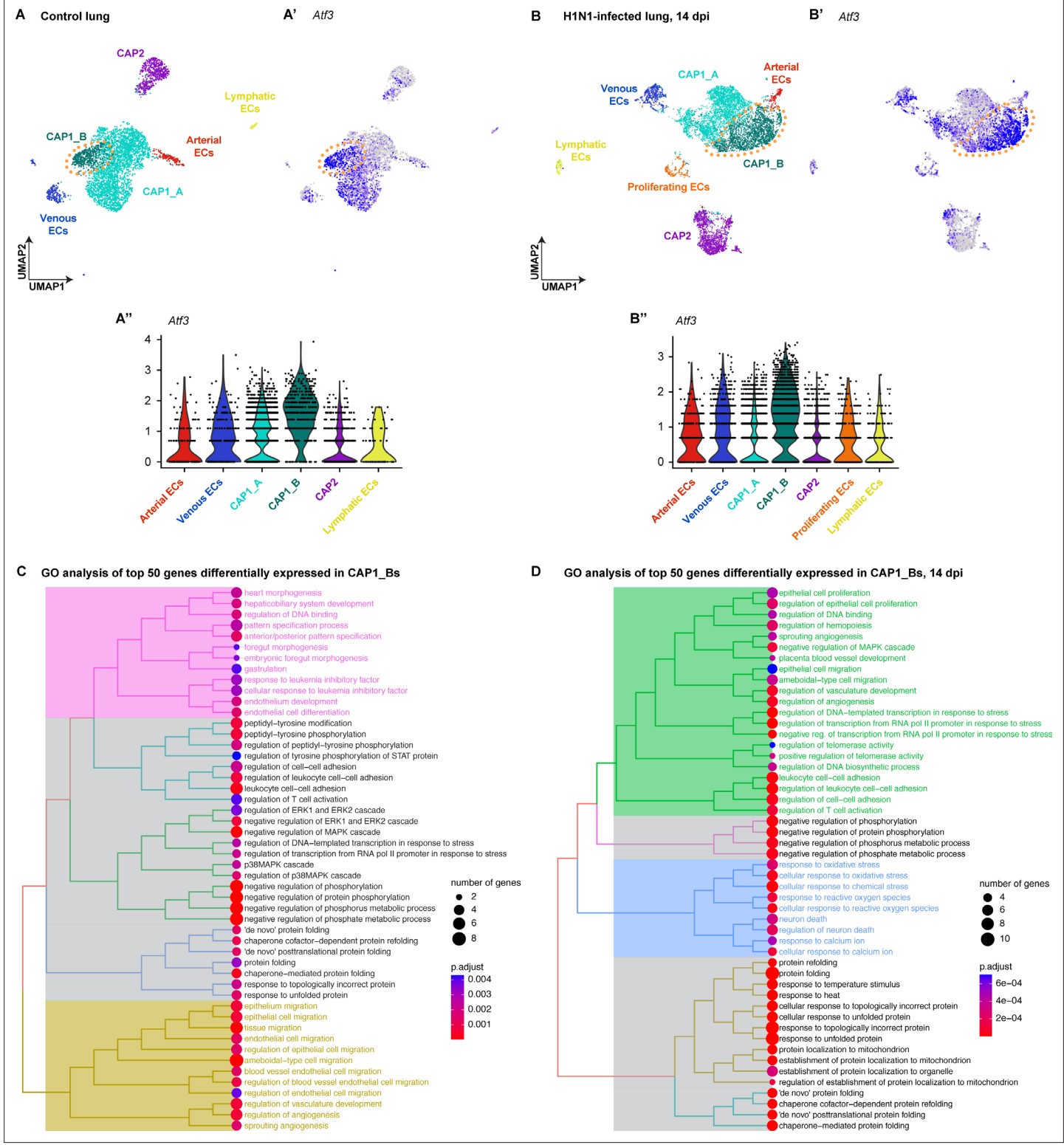

**Figure 1.** Expression of *Atf3* defines a subset of pulmonary capillary endothelial cells. (**A**) The Uniform Manifold Projection (UMAP) of endothelial cell subtypes from single-cell RNA sequencing (scRNA-seq) of whole mouse lung at homeostasis (n=1). (**A', A''**) *Atf3* transcript expression in endothelial cell clusters at homeostasis. Purple represents the high expression of *Atf3*, while gray represents low or no expression. Orange dotted line outlines the CAP1_B subcluster defined by the high expression of *Atf3*. (**B**) UMAP projection of endothelial cell subtypes from scRNA-seq of whole mouse lung at 14 days post-H1N1 influenza infection (dpi) (n=2). (**B', B''**) *Atf3* transcript expression in endothelial cell clusters at 14 dpi. Purple represents the high expression of *Atf3*, while gray represents low or no expression. The number of cells in the CAP1_B cluster is increased at 14 dpi compared to

*Figure 1 continued on next page*

*Figure 1 continued*

the control lung. Orange dotted line outlines the CAP1_B subcluster defined by the high expression of *Atf3*. (**C**) Tree plot of the top fifty differentially expressed genes in CAP1_Bs at homeostasis. CAP1_Bs are defined by the expression of genes involved in endothelial development and differentiation, protein phosphorylation, cell-cell adhesion, protein folding, MAPK signaling, endothelial cell migration, and angiogenesis. (**D**) Tree plot of the top fifty differentially expressed genes in CAP1_Bs at 14 days post influenza infection. CAP1_Bs are defined by some of the same gene ontology (GO) terms during regeneration as they are at homeostasis, including cell-cell adhesion, protein phosphorylation, and protein folding. However, CAP1_Bs after H1N1 are also defined by the expression of genes involved in angiogenesis, vascular development, and cellular response to stress. This reveals a putative role for *Atf3*-expressing CAP1_Bs in the regenerative response of the endothelium to influenza infection.

The online version of this article includes the following figure supplement(s) for figure 1:

**Figure supplement 1.** Pulmonary endothelial cell (EC) subclusters at homeostasis are defined by their differentially expressed genes.

**Figure supplement 2.** Pulmonary endothelial cell (EC) subclusters at 14 days post-H1N1 are defined by their differentially expressed genes.

**Figure supplement 3.** Integration of control and influenza single-cell RNA sequencing (scRNA-seq) data.

---

state continues to express genes involved in vascular development and angiogenesis, but also genes involved in cellular stress response (*Figure 1D*). This indicates that the *Atf3*-expressing CAP1_B subcluster at 14 dpi likely represents a group of cells responding to acute injury. The existence of this population at homeostasis suggests that *Atf3*-expressing CAP1_Bs are involved in the maintenance of homeostasis or exist in a primed state to respond to acute injury.

## Atf3-expressing endothelial cells increase in number after lung injury

To define the functional relevance of *Atf3* expression in pulmonary capillary ECs after acute lung injury and determine whether *Atf3*-expressing CAP1_Bs preferentially respond to viral lung injury, we crossed a mouse line expressing *CreERT2* under the control of the endogenous *Atf3* promoter (*Atf3^CreERT2^*) (*Denk et al., 2015*; *Holland et al., 2019*) to a *ROSA26^LSL-tdTomato^* reporter allele (*Madisen et al., 2010*) to perform cell type-specific lineage tracing in response to acute lung injury. *Atf3^CreERT2^*; *ROSA26^LSL-tdTomato^* (Atf3-tdTm) mice were injured using intranasal administration of the H1N1 PR8 influenza virus or PBS control, leading to a spatially heterogeneous injury similar to human influenza (*Kumar et al., 2011*; *Liberti et al., 2021*; *Töpfer et al., 2014*; *Zacharias et al., 2018*). To assess the role of injury-responsive Atf3-expressing cells in the distal lung, we treated mice with tamoxifen at 8 days post-injury (dpi). Mice received EdU in their drinking water immediately after tamoxifen at 8 dpi until 14 dpi, a post-H1N1 time window with increased endothelial proliferation (*Niethamer et al., 2020*). Lungs were analyzed at 3 weeks after injury (*Figure 2A*). To assess the role of ATF3-expressing cells at homeostasis, we treated mice with tamoxifen to label cells two weeks before injury. Mice received EdU in their drinking water from 14 to 22 dpi, a post-H1N1 time window with the highest endothelial proliferation (*Niethamer et al., 2020*). Lungs were analyzed at 3 weeks after injury (*Figure 2B*). In contrast with reports in the brain (*Holland et al., 2019*), injury to the distal lung did not result in tamoxifen-independent Cre recombination (*Figure 2—figure supplement 1A, B*). Consistent with transcript expression in our scRNA-seq data, we observed Atf3-tdTm expression in pulmonary ECs, measured by flow cytometry for CD45⁻/tdTm⁺/CD31⁺ cells, at homeostasis in PBS-treated mice (*Figure 2C*; *Figure 2—figure supplement 1C*). In mice induced after H1N1 injury, however, the percentage of Atf3-positive pulmonary ECs in the distal lung was dramatically increased (*Figure 2C*; *Figure 2—figure supplement 1D*). Although *Atf3* expression has been reported in proliferating AT2 cells in mice infected with *Streptococcus pneumoniae* (*Ali et al., 2022*), we did not observe an increase in Atf3-positive AT2 cells or in Atf3-positive mesenchymal cells following influenza infection (*Figure 2—figure supplement 2A, B*). Our data also show that the increase in Atf3-expressing (CD45⁻/tdTm⁺) cells that we observed after H1N1 was specific to the endothelial compartment, and a similar increase did not occur in CD31⁻ non-ECs (*Figure 2C*, *Figure 2—figure supplement 2C, D*). These data indicate that Atf3 plays a specific role in the pulmonary endothelium during the regenerative response to H1N1 injury.

To determine whether CAP1 proliferation is occurring specifically in Atf3-expressing cells after injury, we analyzed total proliferating ECs in our post-injury induction mice by flow cytometry for CD45⁻/CD31⁺/EdU⁺ cells. This revealed that 40–60% of proliferating ECs between 8–14 dpi express Atf3 ('traced,' tdTm⁺), indicating that Atf3-expressing CAP1_Bs contribute significantly to the EC proliferative response after H1N1 injury, but that not all proliferating ECs express ATF3 (*Figure 2D*; *Figure 2—figure supplement 1E, F*). To determine whether *Atf3* expression at homeostasis defines

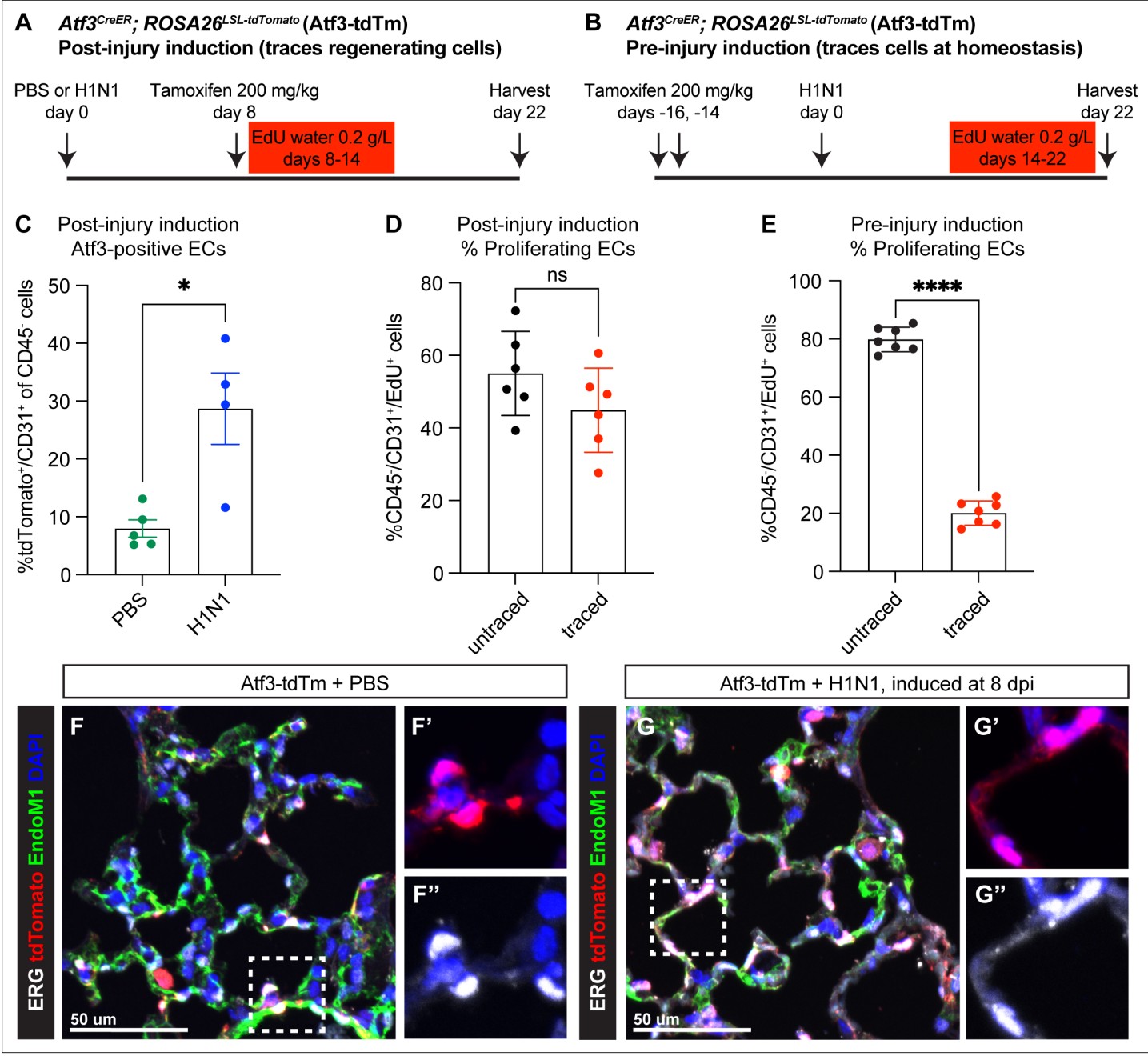

**Figure 2.** *Atf3*-expressing CAP1_Bs increase in the mouse lung after H1N1 injury and contribute to endothelial regeneration. (**A**) Schematic of the experimental setup to investigate *Atf3*-expressing cells during regeneration after H1N1. *Atf3*$^{CreERT2}$; *ROSA26*$^{LSL-tdTomato}$ (Atf3-tdTm) adult mice (5–13 weeks old) received 200 mg/kg tamoxifen by oral gavage 8 days after H1N1 influenza injury (8 dpi) and were treated with EdU ad libitum in their drinking water from 8 to 14 dpi. Mice were harvested at 22 dpi for flow cytometry and immunofluorescence (IF) analysis. (**B**) Schematic of the experimental setup to investigate cells that express *Atf3* at homeostasis. Atf3-tdTm adult mice (8–13 weeks old) received 200 mg/kg tamoxifen by oral gavage at 16 and 14 days prior to H1N1 infection. Following H1N1 administration, mice were treated with EdU ad libitum in their drinking water from 14 to 22 dpi and harvested at 22 dpi. (**C**) A large increase in Atf3-positive ECs (CD31$^+$/tdTomato$^+$ cells) is observed in Atf3-tdTm mice infected with H1N1 (n=4) compared to control mice (n=5). Error bars represent standard error of the mean (SEM). *p=0.0317 by Mann-Whitney rank-sum test. (**D**) Percentage of proliferating ECs (CD45$^−$/CD31$^+$/EdU$^+$ cells) between 8–14 dpi represented by Atf3-positive (traced) and Atf3-negative (untraced) cells when Cre activity is induced during regeneration (n=6 animals). This demonstrates that Atf3-positive ECs are not the sole source of proliferating ECs during regeneration. Error bars represent standard deviation (SD). ns, p=0.1604 by unpaired t-test. (**E**) Percentage of proliferating ECs (CD45$^−$/CD31$^+$/EdU$^+$) derived from ECs that were Atf3-negative at homeostasis (untraced) versus Atf3-positive at homeostasis (traced) (n=6 animals). While traced cells clearly contribute to endothelial proliferation after H1N1, untraced cells make up approximately 80% of the proliferating endothelial cells (ECs) between days 14–22. Error bars represent standard deviation (SD). ****p<0.0001, unpaired t-test. (**F**) IF for tdTomato, the nuclear endothelial marker ERG, and the cytoplasmic endothelial marker

*Figure 2 continued on next page*

*Figure 2 continued*

Endomucin-1 in uninjured Atf3-tdTm mice show that Atf3-positive ECs can be found in both large vessels and in the alveolar space at homeostasis (inset marked by white box). Scale bar, 50 µm. (**G**) IF for tdTomato, ERG, and Endomucin-1 in H1N1-infected Atf3-tdTm mice at 22 dpi demonstrates an increase in Atf3-positive ECs in the alveolar space after influenza (inset marked by white box). Scale bar, 50 µm.

The online version of this article includes the following figure supplement(s) for figure 2:

**Figure supplement 1.** *Atf3$^{CreERT2}$* activity is not induced by H1N1 injury.

**Figure supplement 2.** *Atf3* is expressed at low levels in other alveolar cell types, but these cells do not increase in number after H1N1.

**Figure supplement 3.** Atf3-expressing endothelial cells increase in the alveolar space following H1N1 infection.

a population of CAP1s primed to proliferate after injury, we analyzed total proliferating ECs in our pre-injury induction mice. We found that both tdTm$^+$ (Atf3 lineage, 'traced') and tdTm$^-$ (non-Atf3-lineage, 'untraced') CAP1s proliferate, and cells that express *Atf3* at homeostasis contribute to proliferation in a proportion that correlates to the fraction of CAP1s expressing *Atf3* at homeostasis (~20%, *Figure 2E*; *Figure 2—figure supplement 1G, H*). This indicates that *Atf3* expression at homeostasis does not preferentially prime ECs for proliferation after injury. Analysis of the spatial distribution of tdTomato$^+$ cells in the distal lung revealed ATF3-expressing ECs in both large vessels and in the alveolar space at homeostasis (*Figure 2F*; *Figure 2—figure supplement 3A*). However, the increase in Atf3-expressing endothelial cells was observed primarily in the alveolar space (*Figure 2G*; *Figure 2—figure supplement 3B*). These data indicate that both *Atf3*-expressing ECs and non-expressing ECs proliferate after injury.

## Endothelial loss of Atf3 leads to defects in alveolar regeneration

To determine the specific role of endothelial Atf3 in tissue regeneration after acute lung injury, we used the endothelial-specific CreERT2-expressing mouse line *Cdh5-CreERT2* (*Sörensen et al., 2009*) to both trace ECs with the *ROSA26$^{LSL-tdTomato}$* reporter allele and delete *Atf3* using a floxed allele (*Wolford et al., 2013*). *Cdh5$^{CreERT2}$; ROSA26$^{LSL-tdTomato}$; Atf3$^{lox/lox}$* (Atf3$^{EC-KO}$) and control *Cdh5$^{CreERT2}$; ROSA26$^{LSL-tdTomato}$* (Atf3$^{EC-WT}$) mice were subjected to influenza injury and assessed at 14 and 21 dpi for PDPN expression to define alveolar architecture (*Figure 3A*, *Figure 3—figure supplement 1*). Influenza-injured Atf3$^{EC-KO}$ mice had similar amounts of tissue damage and inflammation at 21 dpi when compared to Atf3$^{EC-WT}$ mice (*Figure 3—figure supplement 2*). However, we found that compared to control Atf3$^{EC-WT}$ mice, Atf3$^{EC-KO}$ mice have an abnormal alveolar structure in non-inflamed regions of lung tissue characterized by enlarged airspaces and increased alveolar area at 14 dpi (*Figure 3B–D*). This defect persisted at 21 dpi, indicating that the capacity for regeneration of the distal lung tissue structure is compromised when Atf3 is lost in the pulmonary endothelium (*Figure 3E–G*). Notably, although *Atf3* is also expressed in a subset of CAP1s at homeostasis, we did not observe a decrease in tdTomato$^+$ ECs or a significant change in alveolar cell number or alveolar structure in Atf3$^{EC-KO}$ mice compared to Atf3$^{EC-WT}$ mice in the absence of lung injury at one month or any difference in alveolar area at six months after tamoxifen induction (*Figure 3—figure supplement 3* and *Figure 3—figure supplement 4*). These data indicate that the most critical role of Atf3 in pulmonary ECs is in the response to acute lung injury, and not in homeostasis of the alveolus.

To determine downstream genes and pathways that are affected by the loss of endothelial Atf3 during lung regeneration, we isolated pulmonary ECs from Atf3$^{EC-WT}$ and Atf3$^{EC-KO}$ mice using FACS for tdTomato$^+$ cells at 14 days post influenza infection and performed population RNA sequencing to determine gene expression differences in Atf3-KO ECs during this key phase of injury repair (*Figure 4A*). Atf3-WT and -KO ECs clustered separately in a principal components analysis, with the loss of Atf3 defining a majority of the total variance in the dataset (*Figure 4B*). Analysis of specific genes that are differentially expressed between groups showed that loss of *Atf3* expression in pulmonary ECs during regeneration results in the downregulation of pathway components of several signaling axes associated with cell proliferation, differentiation, or angiogenesis, including FGF (*Fgfr1*), Notch (*Notch4*), Wnt (*Lrp5, Fzd1*), and *Map3k6*, an activator of the JNK pathway that regulates VEGF expression and signaling (*Figure 4C and D*; *Figure 4—figure supplement 1*). In addition, Atf3-KO ECs demonstrated downregulation of genes that are known or suggested regulators of cytoskeletal organization, such as *Limk1, Sdc3, Pld2, Prkci*, and the small GTPases *Rab5a, Rab10*, and *Rab35* (*Figure 4C and D*; *Figure 4—figure supplement 1*). These data support our scRNA-seq analysis indicating that

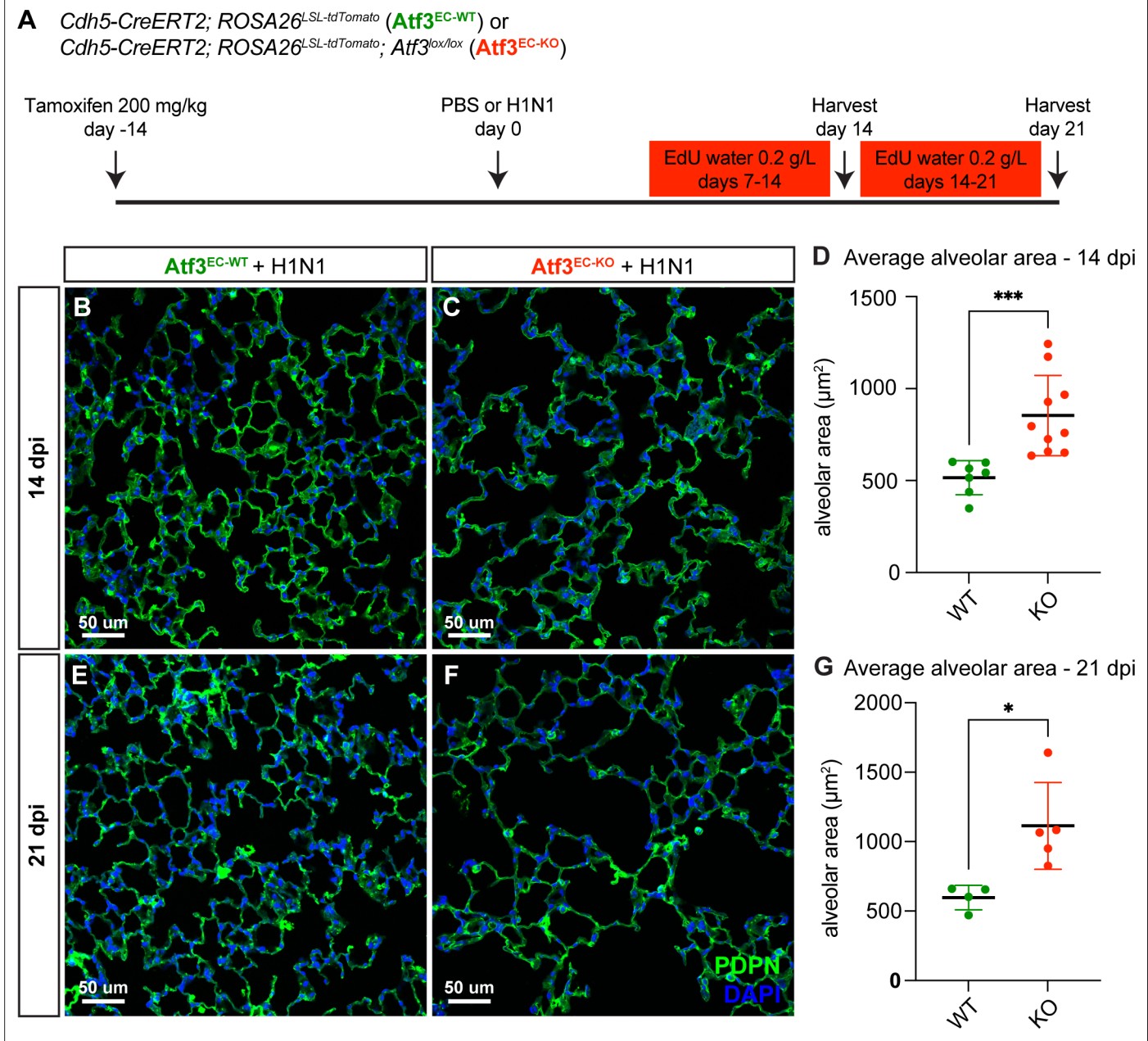

**Figure 3.** Endothelial loss of Atf3 causes defects in alveolar regeneration. (**A**) Schematic of the experimental setup. *Cdh5-CreERT2; ROSA26*[LSL-tdTomato] (Atf3[EC–WT]) and *Cdh5-CreERT2; ROSA26*[LSL-tdTomato]*; Atf3*[lox/lox] (Atf3[EC-KO]) adult mice (5–15 weeks old) received 200 mg/kg tamoxifen by oral gavage 2 weeks before H1N1 influenza injury or PBS administration (control). Mice received EdU in their drinking water at 0.2 g/L from days 7–14 and were harvested at 14 dpi or received EdU in their drinking water at 0.2 g/L from days 14–21 and were harvested at 21 dpi. (**B**) Representative image of immunofluorescence (IF) for alveolar epithelial type 1 (AT1) cell marker podoplanin (PDPN), used to visualize alveolar structure across the tissue, in an Atf3[EC-WT] mouse at 14 dpi. (**C**) Representative image of IF for PDPN in an Atf3[EC-KO] mouse at 14 dpi, demonstrating an increase in alveolar area compared to the WT mouse. (**D**) Mice that have lost endothelial Atf3 expression (n=10) have significantly increased alveolar area at 14 dpi compared to Atf3[EC-WT] mice (n=7). Error bars represent standard deviation (SD). ***p=0.0001 by Mann-Whitney rank-sum test. (**E**) Representative image of IF for PDPN to define alveolar space in an Atf3[EC-WT] mouse at 21 dpi. (**F**) Representative image of IF for PDPN to define alveolar space in an Atf3[EC-KO] mouse at 21 dpi, demonstrating an increase in alveolar area compared to the WT mouse. (**G**) Mice with endothelial Atf3 knockout (n=5) have significantly increased alveolar area at 21 dpi compared to WT mice (n=4). Error bars represent standard deviation (SD). *p=0.0159 by Mann-Whitney rank-sum test. Scale bars, 50 μm.

The online version of this article includes the following figure supplement(s) for figure 3:

**Figure supplement 1.** Quantification of the alveolar area using immunohistochemistry for podoplanin (PDPN).

**Figure supplement 2.** Atf3[EC-WT] and Atf3[EC-KO] animals have similar levels of tissue damage after influenza infection.

*Figure 3 continued on next page*

**Figure supplement 3.** Endothelial loss of Atf3 at homeostasis does not cause defects in lung tissue structure or cell loss after 28 days.

**Figure supplement 4.** Loss of endothelial Atf3 at homeostasis does not cause tissue structure defects after 6 months.

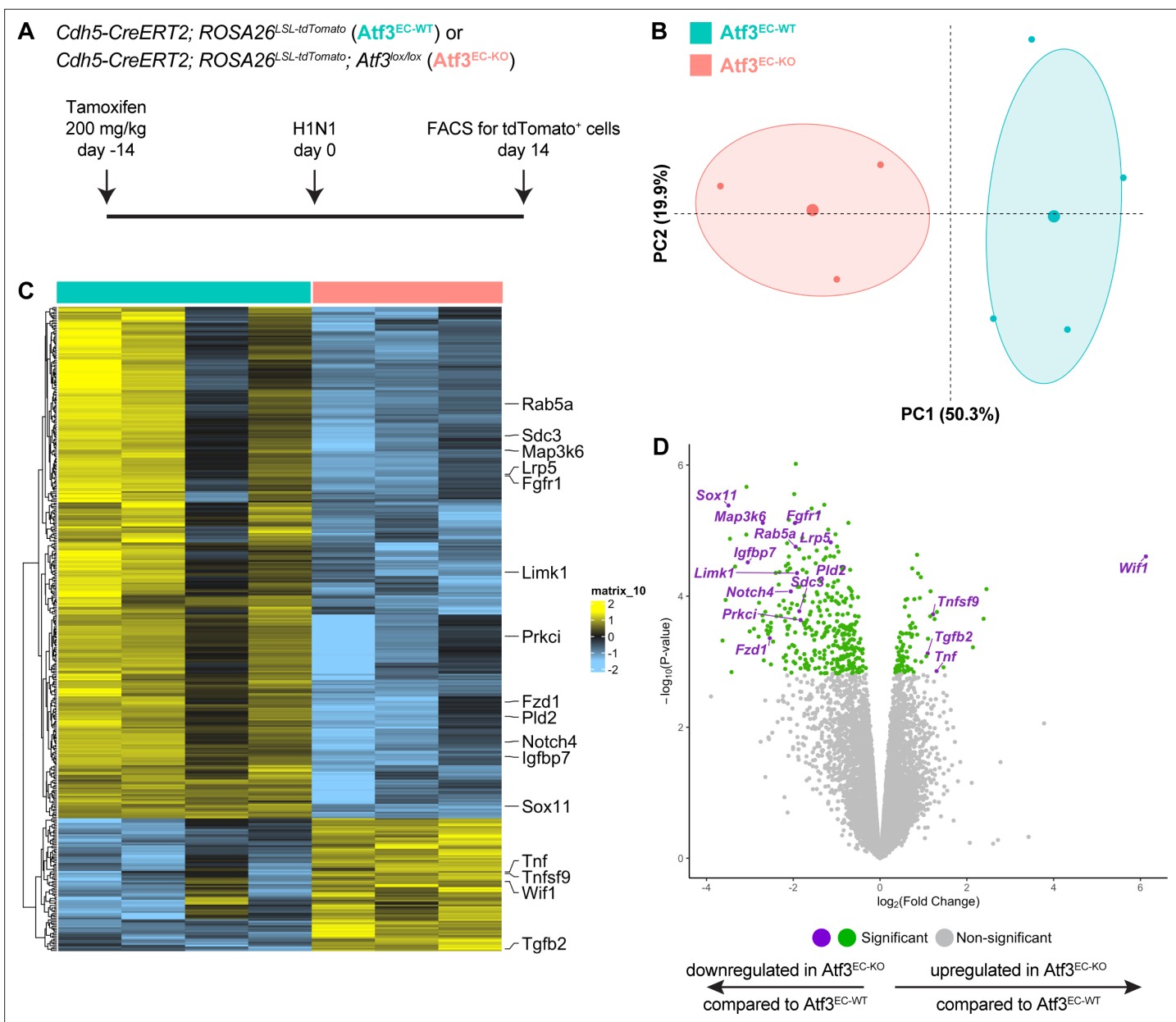

**Figure 4.** Transcriptional changes due to loss of Atf3 after acute influenza injury. (**A**) Schematic of the experimental setup. *Cdh5-CreERT2; ROSA26<sup>LSL-tdTomato</sup>* (Atf3<sup>EC-WT</sup>, n=4) and *Cdh5-CreERT2; ROSA26<sup>LSL-tdTomato</sup>; Atf3<sup>lox/lox</sup>* (Atf3<sup>EC-KO</sup>, n=3) adult mice received 200 mg/kg tamoxifen by oral gavage 2 weeks before H1N1 influenza injury. At 14 dpi, endothelial cells (ECs) were isolated from WT and KO mice using FACS for tdTomato-positive cells. (**B**) Principal components analysis demonstrates that the expression of Atf3 defines a majority of the total variance in the dataset. (**C**) Loss of Atf3 expression in ECs results in more downregulated genes than upregulated genes. Downregulated genes include transcription factors, cytoskeletal genes, and members of the Fgf, Wnt, and Notch signaling pathways. Upregulated genes include members of the Tnf family of cytokines, the Wnt inhibitor *Wif1*, and *Tgfb2*. (**D**) Volcano plot highlighting significantly upregulated and downregulated genes in Atf3<sup>EC-KO</sup> compared to Atf3<sup>EC-WT</sup> animals.

The online version of this article includes the following figure supplement(s) for figure 4:

**Figure supplement 1.** qRT-PCR for expression of *Atf3* and downstream genes in Atf3<sup>EC-WT</sup> and Atf3<sup>EC-KO</sup> animals.

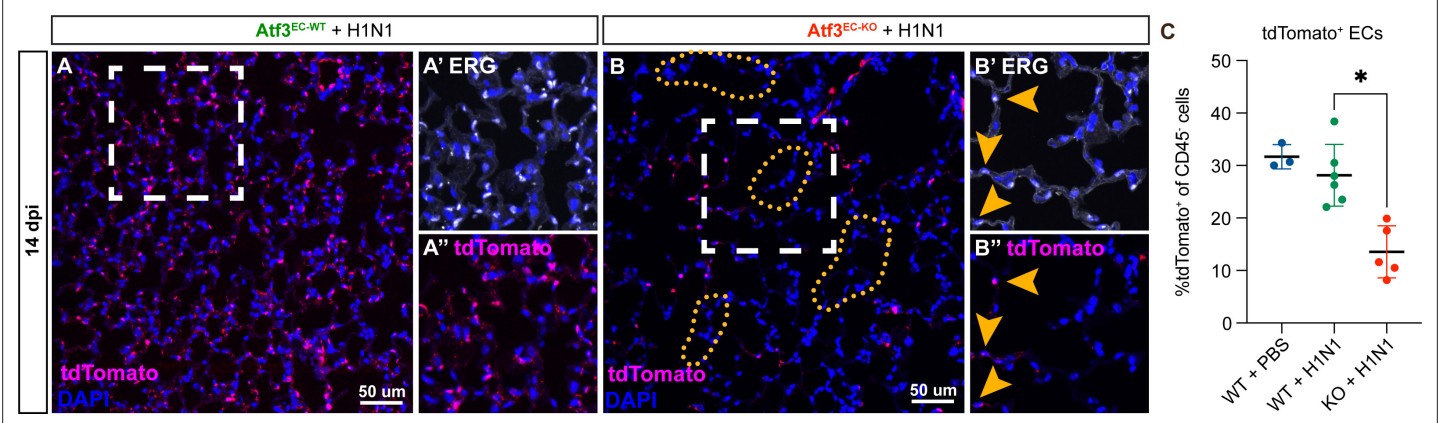

**Figure 5.** Endothelial loss of Atf3 causes alveolar endothelial cell loss corresponding to regions of altered alveolar architecture. (**A**) Immunofluorescence (IF) for tdTomato (magenta) and DAPI (blue) at 14 days post-H1N1 shows an expected distribution of tdTomato⁺ ECs throughout the alveolar space in Atf3^EC-WT animals. (**A'**) Higher magnification image of the area within the white box in (**A**) shows IF for the endothelial marker ERG (white). (**A"**) Higher magnification image of the area within the white box in (**A**) shows IF for tdTomato (magenta). (**B**) Atf3^EC-KO mice at 14 dpi demonstrate heterogeneous loss of tdTomato⁺ endothelial cells (ECs). Regions with increased alveolar area correlate with regions devoid of tdTomato⁺ cells (yellow dotted lines). (**B'**) Higher magnification image of the area within the white box in (**B**) shows IF for the endothelial marker ERG (white). Yellow arrows indicate the few remaining ERG⁺ ECs. (**B"**) Higher magnification image of the area within the white box in (**B**) showing IF for tdtomato (magenta). Yellow arrows indicate the few remaining tdTomato⁺ ECs. (**C**) At 14 days post-H1N1, Atf3^EC-KO mice (n=5) show a significant decrease in tdTomato-expressing cells, indicating a loss of ECs, compared to PBS-treated (n=3) or H1N1-treated (n=6) Atf3^EC-WT mice. Error bars represent standard deviation (SD). *p=0.0447 using Dunn's multiple comparisons test following a Kruskal-Wallis test (p=0.0008). Scale bars, 50 μm.

The online version of this article includes the following figure supplement(s) for figure 5:

**Figure supplement 1.** Endothelial cell loss in regions of altered alveolar architecture persists at 21 days post-H1N1.

**Figure supplement 2.** Endothelial loss of Atf3 leads to loss of alveolar epithelial type 1 (AT1) and AT2 cells but not alteration in AT2-AT1 differentiation.

**Figure supplement 3.** Endothelial loss of Atf3 does not impair CAP1-CAP2 differentiation.

the *Atf3*-expressing CAP1_B population expresses genes involved in angiogenesis, vascular development, and cell migration (*Figure 1D*). Among the group of signaling molecules upregulated in Atf3-KO ECs were the TGFB signaling family member *Tgfb2* and the Wnt signaling pathway inhibitor *Wif1*. GO analysis using GAGE demonstrated that cytokine activity (GO:0005125) is also upregulated in Atf3-KO ECs, as evidenced by increased expression of *Tnf*, *Tnfsf9*, *Clcf1*, and *Bmp4*. These data support our scRNA-seq analysis indicating that CAP1_Bs are involved in cellular stress response and suggest they respond to injury by increasing cytokine signaling. Loss of Atf3 in CAP1s may, therefore, affect their ability both to respond to proliferative and regenerative cues from other alveolar cell types and conversely to communicate with other cell types in the alveolus to promote tissue regeneration after infectious injury.

## Altered alveolar architecture in Atf3 EC-KO animals due to loss of capillary endothelial cells

The structural defects in Atf3^EC-KO animals suggested an inability of Atf3-KO CAP1s to respond to regenerative signals that are required to initiate EC proliferation and promote vascular repair after injury. To explore these defects in more detail, we examined the spatial distribution of the ERG⁺/tdTomato⁺ pulmonary capillary ECs in the alveolar space of Atf3^EC-WT and Atf3^EC-KO mice. At 14 days post-influenza infection, Atf3^EC-WT animals have an expected distribution of pulmonary ECs throughout the alveolar space (*Figure 5A*). In contrast, Atf3^EC-KO animals demonstrate the patchy distribution of pulmonary ECs, with large regions only sparsely populated with tdTomato⁺/ERG⁺ cells (*Figure 5B*; yellow arrows in **B'** and **B"**). Furthermore, we observed that loss of alveolar architecture correlates with regions lacking pulmonary ECs (*Figure 5B*, yellow dotted lines in **B** and **B'**). These defects persist at 21 dpi (*Figure 5—figure supplement 1*). We quantified the total pulmonary EC number and found no difference in the number of tdTomato⁺ cells between PBS-treated and H1N1-treated Atf3^EC-WT animals (*Figure 5C*). However, Atf3^EC-KO mice showed a significant decrease in tdTomato⁺ cells at 14 dpi (*Figure 5C*). These data indicate that Atf3 is required to maintain the proper number of pulmonary

ECs in the lung alveolus after acute injury and that loss of endothelium in Atf3$^{EC-KO}$ mice leads to defective alveolar regeneration.

To determine whether loss of vasculature and loss of structure in Atf3$^{EC-KO}$ mice affects epithelial regeneration, we compared IF for the AT2 cell marker Lamp3, AT1 cell marker Hopx, and a total epithelial marker Nkx2.1 in Atf3$^{EC-WT}$ and Atf3$^{EC-KO}$ animals at 21 days post-H1N1 (*Figure 5—figure supplement 2A, B*). We observed a decrease in the total number of AT1 and AT2 cells, consistent with the loss of alveolar structure in Atf3$^{EC-KO}$ animals and indicating that defective endothelial regeneration leads to a loss of alveolar epithelium (*Figure 5—figure supplement 2C, D*). However, we found no difference in the ratio of AT1 to AT2 cells between Atf3$^{EC-WT}$ and Atf3$^{EC-KO}$ animals, indicating that Atf3-deficient endothelium does not lead to defective AT2-AT1 differentiation (*Figure 5—figure supplement 2E*). To determine whether loss of Atf3 in the vasculature affects endothelial differentiation, we compared the number of *Car4*$^+$ CAP2s between Atf3$^{EC-WT}$ and Atf3$^{EC-KO}$ mice (*Figure 5—figure supplement 3A, B*). While we observed fewer CAP2s, the ratio of CAP2s to total endothelium was unchanged, suggesting that the loss of CAP1s does not affect significantly affect the differentiation of the remaining CAP1s into CAP2s (*Figure 5—figure supplement 3C*).

## Loss of *Atf3* leads to decreased proliferation and increased apoptosis in pulmonary ECs

We next examined whether there were defects in EC proliferation or survival after the injury that could explain the loss of alveolar architecture observed in Atf3$^{EC-KO}$ animals subjected to influenza injury. We quantified EC proliferation in the second and third weeks after H1N1 infection by flow cytometry for CD45$^-$/CD31$^+$/EdU$^+$ cells. Between 7 and 14 dpi, Atf3$^{EC-KO}$ animals show a significant decrease in the percentage of proliferating ECs in comparison to Atf3$^{EC-WT}$ animals (*Figure 6A*). This defect persists between 14 and 21 dpi, indicating that loss of endothelial Atf3 diminishes the ability of the endothelium to proliferate during the regenerative response to acute injury (*Figure 6B*). Because we found that Atf3$^{EC-KO}$ ECs have increased cytokine activity and proinflammatory cytokines can activate apoptosis in other contexts (*Grunnet et al., 2009*), we also examined cell death in Atf3$^{EC-KO}$ animals. Quantification of apoptosis by IF for cleaved caspase 3 at 14 dpi revealed a striking increase in the number of caspase-3$^+$/tdTomato$^+$ cells in Atf3$^{EC-KO}$ compared to Atf3$^{EC-WT}$ animals, indicating that ATF3$^{EC-KO}$ pulmonary ECs are indeed more sensitive to apoptosis during this regenerative period in the endothelium (*Figure 6C–E*). Taken together, these data suggest that loss of endothelium and associated loss of alveolar structure in Atf3$^{EC-KO}$ mice occurs both through a decrease in the ability of Atf3-KO ECs to respond to proliferative signals and an increase in sensitivity to apoptosis after acute injury to the distal lung.

To determine whether loss of proliferative response and increased apoptosis have a detrimental impact on long-term regeneration, we compared the tissue structure of Atf3$^{EC-KO}$ and Atf3$^{EC-WT}$ animals by IF for PDPN at 90 dpi (*Figure 7A*). Compared to Atf3$^{EC-WT}$ mice, Atf3$^{EC-KO}$ mice have persistent alveolar architectural defects at 90 dpi (*Figure 7B–F*). The variability in these differences likely reflects the heterogenous response to acute injury previously reported in the lung (*Liberti et al., 2021*; *Zacharias et al., 2018*) and the ability for residual Atf3-replete pulmonary ECs to repopulate some regions. Despite this range of responses, Atf3$^{EC-KO}$ animals had statistically increased airspace area at 90 dpi, indicating persistent abnormalities after injury upon loss of Atf3 (*Figure 7G*). These data reveal Atf3 as an essential mediator of endothelial regeneration following acute lung injury and that loss of Atf3 leads to persistent defects in alveolar architecture accompanied by loss of AT1 and AT2 cells.

## Discussion

Endothelial proliferation increases following various lung injuries and appears critical for vascular regeneration (*Gillich et al., 2020*; *Niethamer et al., 2020*). In this study, we sought to determine factors that distinguish CAP1 and CAP2 pulmonary ECs in the distal lung and that permit or promote CAP1 proliferation after injury. Using single-cell transcriptomic analysis, we identified the transcription factor Atf3 as a putative marker of a previously unrecognized CAP1 endothelial subpopulation whose gene expression suggests that it is a critical regulator of the regenerative response by pulmonary ECs. We show that loss of *Atf3* expression in the endothelium diminished the regenerative capacity of the distal lung, resulting in a persistent emphysema-like phenotype and decreasing the activity of

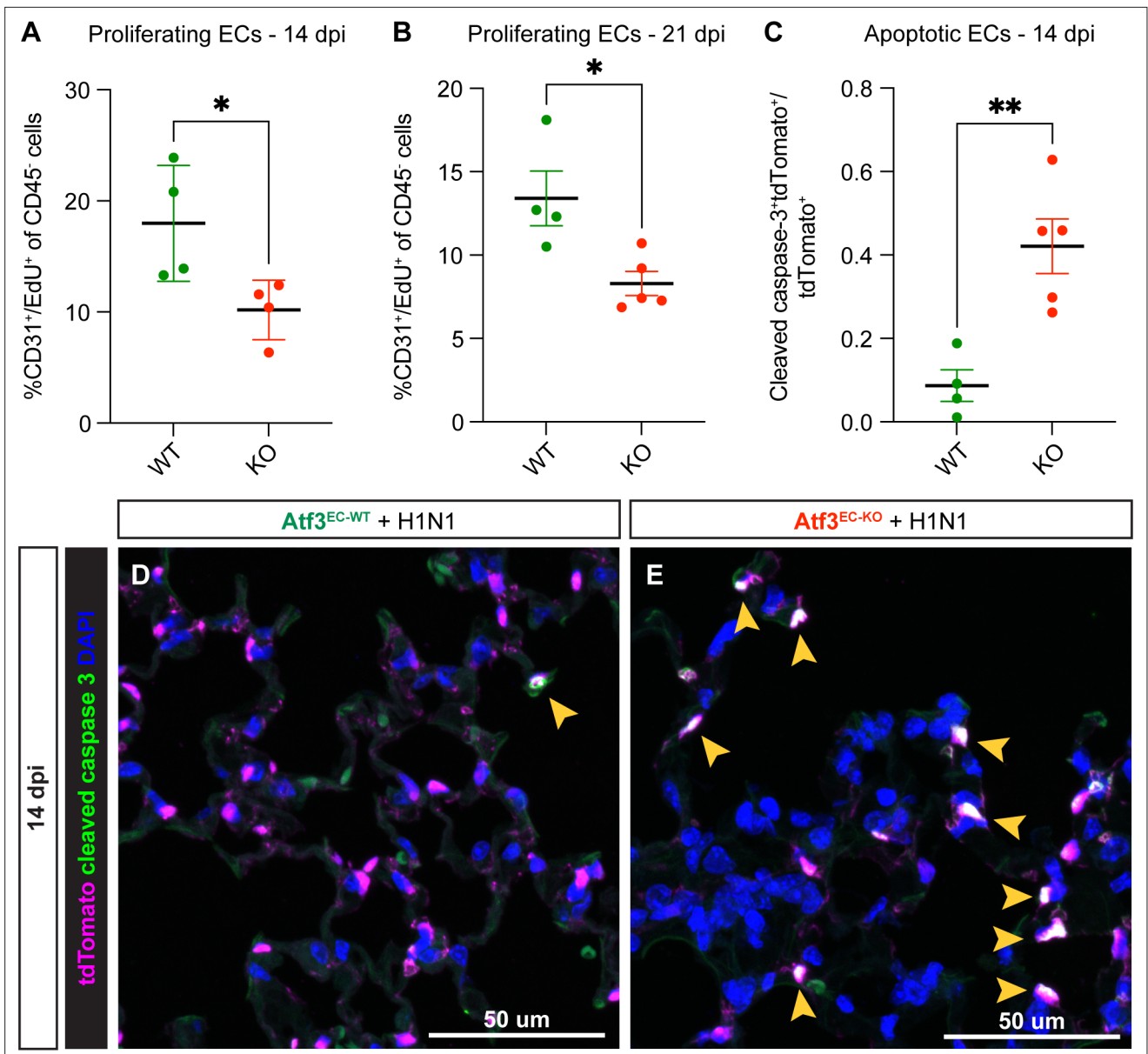

**Figure 6.** Atf3 loss in endothelial cells (ECs) impairs the regenerative response to H1N1. (**A**) Compared to Atf3$^{EC-WT}$ mice (n=4), Atf3$^{EC-KO}$ mice (n=4) have significantly decreased EC proliferation (%CD31$^+$/EdU$^+$ of CD45$^-$ cells) between 7–14 dpi. Error bars represent standard deviation (SD). *p=0.0377 by unpaired t-test. (**B**) Compared to Atf3$^{EC-WT}$ mice (n=4), Atf3$^{EC-KO}$ mice (n=5) have significantly decreased EC proliferation (%CD31$^+$/EdU$^+$ of CD45$^-$ cells) between 14–21 dpi, the time period during which EC proliferation peaks after H1N1. Error bars represent standard error of the mean (SEM). *p=0.0177 by unpaired t-test. (**C**) Compared to Atf3$^{EC-WT}$ mice (n=4), Atf3$^{EC-KO}$ mice (n=5) demonstrate significantly increased EC apoptosis as measured by cleaved caspase-3$^+$/tdTomato$^+$ endothelial cells as a fraction of all tdTomato$^+$ ECs. Error bars represent standard error of the mean (SEM). **p=0.0045 by unpaired t-test. (**D**) Relatively few cleaved caspase-3$^+$/tdTomato$^+$ apoptotic ECs (yellow arrow) are visible in lung tissue of Atf3$^{EC-WT}$ mice at 14 dpi. (**E**) In contrast, many of the tdTomato$^+$ ECs in Atf3$^{EC-KO}$ mice are cleaved caspase-3$^+$ (yellow arrows). Scale bars in (**D**), (**E**), 50 μm.

several signaling pathways known to be important for angiogenesis while increasing cytokine expression. Endothelial Atf3 knockout mice had decreased numbers of pulmonary ECs caused by decreased proliferation and increased apoptosis, diminishing the ability of pulmonary ECs to respond to stress and impairing their ability to mount a regenerative response. This loss of pulmonary ECs resulted in a concomitant loss of alveolar AT1 and AT2 cells, leading to long-term architectural defects in endothelial Atf3 null mice. These results reveal the necessity of proper lung capillary regeneration after acute injury and the essential role that Atf3 plays in this process.

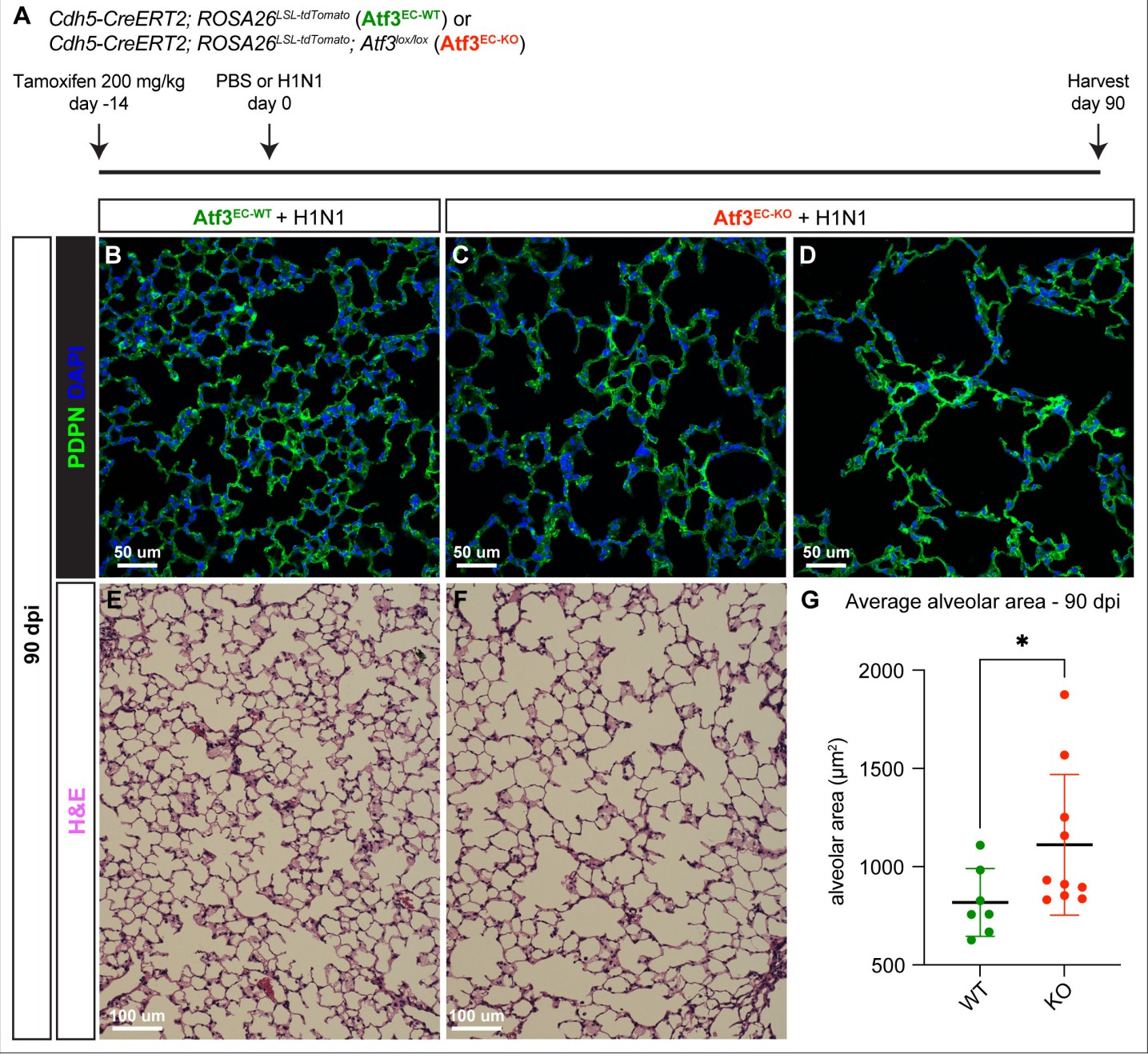

**Figure 7.** Loss of Atf3 in endothelial cells (ECs) impairs long-term alveolar tissue regeneration. (**A**) Schematic of the experimental setup. *Cdh5-CreERT2; ROSA26^LSL-tdTomato* (Atf3^EC-WT^) and *Cdh5-CreERT2; ROSA26^LSL-tdTomato^; Atf3^lox/lox^* (Atf3^EC-KO^) adult mice received 200 mg/kg tamoxifen by oral gavage 2 weeks before H1N1 influenza injury. At 90 dpi, lung tissue from WT and KO mice was collected for immunofluorescence (IF) analysis. (**B**) Representative image of an Atf3^EC-WT^ mouse lung at 90 days post-H1N1 infection. The AT1 cell marker podoplanin (PDPN) was used to visualize alveolar structure across the tissue. (**C**) Representative image of an Atf3^EC-KO^ mouse lung at 90 dpi demonstrates that loss of alveolar structure persists through regeneration. (**D**) Although the lung tissue of several Atf3^EC-KO^ animals appeared as in (**C**), several had more extreme defects, with almost total loss of normal alveolar structure. (**E**) Representative lung tissue section of an Atf3^EC-WT^ mouse at 90 days post-H1N1 infection, stained with hematoxylin and eosin. (**F**) Representative image of an Atf3^EC-KO^ mouse lung at 90 dpi, stained with hematoxylin and eosin. (**G**) Mice with endothelial Atf3 knockout (n=10) have significantly increased alveolar area at 90 dpi compared to WT mice (n=7). *p=0.025 by Mann-Whitney rank-sum test. Scale bars in (**B**), (**C**), (**D**), 50 μm. Scale bars in (**E**), (**F**), 100 μm.

After viral lung injury, a delicate balance in inflammatory response must be maintained to clear the virus without causing severe tissue damage. Our data suggest that Atf3 may be one factor that maintains this balance in the lung after an influenza infection. The differential gene expression analysis from our scRNA-seq data suggests that *Atf3*-expressing CAP1_Bs are involved in cellular stress response, while our in vivo data suggests that Atf3-positive CAP1s increase in number after flu injury and that loss of Atf3 leads to structural defects in the alveolus. Further, our RNA-seq data indicate that when Atf3 is lost, multiple signaling molecules that may be important for the CAP1 injury response are downregulated, while cytokine expression is upregulated. Loss of Atf3 in the lung has been found to increase susceptibility to injury in a mouse model of ventilator-induced lung injury through increased inflammatory response, suggesting that Atf3 may normally protect against inflammation (*Akram et al., 2010*). Furthermore, loss of Atf3 in either myeloid cells or parenchymal cells in mouse bone marrow chimeras resulted in increased inflammatory response, with Atf loss in parenchymal cells causing increased cell permeability (*Shan et al., 2015*). Our work indicates that the protective role of Atf3 in the lung extends outside ventilator-induced lung injury to encompass inflammatory injury caused by a viral infection, and we show that the lung endothelium is an essential Atf3-expressing cell that helps to promote alveolar regeneration after infectious injury.

Our data indicate that Atf3 expression is important for both pulmonary EC survival and the CAP1 proliferative response. A previous report suggests that Atf3 can play a dual role in controlling inflammatory response and promoting repair (*Akram et al., 2010*), and the dual increase in apoptosis and loss of proliferation we observe in Atf3-KO ECs suggests that perhaps these cells are both more sensitive to cytokine signaling and less responsive to proliferative cues. These effects may be mediated by one or several signaling pathways downstream of Atf3. Our RNA-seq data indicate that loss of Atf3 results in a decrease in *Notch4* expression, and impaired Notch4 signaling in endothelial cells has been shown to result in EC 'activation' and increases in several inflammatory cytokines in other contexts (*MacKenzie et al., 2004*; *Quillard et al., 2008*). Notch4 has also been suggested to inhibit endothelial apoptosis (*MacKenzie et al., 2004*; *Quillard et al., 2008*), and its loss in Atf3[EC-KO] mice is one possible mechanism for the increase in apoptosis we observe in these animals after H1N1 infection. Our RNA-seq data also demonstrate a decrease in *Fgfr1* expression in Atf3[EC-KO] animals; Fgfr1 signaling has been implicated in neovascularization in the skin and eye, where this pathway plays a role in injury response but not in the maintenance of homeostasis (*Oladipupo et al., 2014*). This suggests that Fgfr1 signaling may act downstream of Atf3 to activate lung EC proliferation and the formation of new vessels after acute lung injury. Notch4 and Fgfr1 represent only two of several putative candidates that may regulate lung endothelial apoptosis and proliferation after H1N1 infection, and the role of these and other pathways in the EC response to influenza merit further evaluation in future studies. In combination, these data suggest that Atf3 is essential for the pulmonary EC regenerative response and that loss of expression causes persistent defects in the alveolar structure after acute injury. The role of Atf3 in regulating inflammatory response suggests that immune-endothelial signaling may play an important role in vascular regeneration after influenza. Future experiments to identify and characterize these potential interactions will help to further elucidate the role of Atf3 in the distal lung.

This work identifies Atf3 as a critical regulator of the pulmonary EC response to acute injury and suggests that this transcription factor may play multiple roles within the endothelium during this response. Our data identify an additional context in which Atf3 acts protectively in cellular stress response, namely, the capillary vasculature in the lung after H1N1 influenza infection. In addition to this function, we find that endothelial expression of *Atf3* is also associated with increasing regenerative response, possibly through promoting angiogenesis after vessels are lost to injury. Finally, Atf3 may act to repair the damaged capillary structure itself through the mediation of pulmonary EC migration or adhesion of epithelial and endothelial cells. With this critical knowledge in hand, identifying and studying the downstream pathways that mediate these separate functions will be essential. Future therapies that target Atf3 or its downstream pathways in pulmonary ECs after viral lung injury may assist in increasing tissue protection from inflammation while permitting appropriate inflammatory response to the virus as well as essential repairs to the capillary vascular structure.

## Materials and methods

### Key resources table

| Reagent type (species) or resource | Designation | Source or reference | Identifiers | Additional information |
|---|---|---|---|---|
| Strain, strain background (*Mus musculus*, male and female) | Atf3<sup>tm1.1(cre/ERT2)Msra</sup> (*Atf3<sup>CreERT2</sup>*) | Matt Ramer | MGI: 6360893 | PMID:26610346 |
| Strain, strain background (*Mus musculus*, male and female) | C57BL/6-Tg(Cdh5-cre/ERT2)1Rha (*Cdh5<sup>CreERT2</sup>*) | Taconic | 13073, MGI: 3848982 | PMID:19144989 |
| Strain, strain background (*Mus musculus*, male and female) | Gt(ROSA)26Sor<sup>tm14(CAG-tdTomato)Hze</sup> (*ROSA26<sup>LSL-tdTomato</sup>*) | Jackson Laboratory | 007914, MGI: 3809524 | PMID:20023653 |
| Strain, strain background (*Mus musculus*, male and female) | Atf3<sup>tm1.1Hai</sup> (*Atf3<sup>lox</sup>*) | Tsonwin Hai | MGI: 5547956 | PMID:23921126 |
| Antibody | tdTomato (goat polyclonal) | Origene | AB8181-200 | 1:50 |
| Antibody | RFP (DsRed) (rabbit polyclonal) | Rockland | 600-401-379, RRID:AB_2209751 | 1:50 |
| Antibody | ERG (mouse monoclonal) | Abcam | ab214341 | 1:100 |
| Antibody | ERG (rabbit monoclonal) | Abcam | ab92513 RRID:AB_2630401 | 1:100 |
| Antibody | Endomucin-1 (goat polyclonal) | R&D Systems | AF4666 RRID:AB_2100035 | 1:500 |
| Antibody | LAMP3 (rat monoclonal) | Novus Biologicals | DDX0191P-100 RRID:AB_2827532 | 1:100 |
| Antibody | HOPX (mouse monoclonal) | Santa Cruz Biotechnology | sc-398703 RRID:AB_2687966 | 1:100 |
| Antibody | NKX2-1 (TTF1) (rabbit monoclonal) | Abcam | ab76013 RRID:AB_1310784 | 1:50 |
| Antibody | PDPN (mouse monoclonal) | Abcam | ab10288 RRID:AB_297027 | 1:100 |
| Antibody | CD45-PerCP-Cy5.5 (rat monoclonal) | Thermo Fisher | 45-0451-82 RRID:AB_1107002 | 1:200 |
| Antibody | CD31-PE-Cy7 (rat monoclonal) | Thermo Fisher | 25-0311-82 RRID:AB_2716949 | 1:100 |
| Chemical compound, drug | PR8-GP33 H1N1 influenza | Dr. E. John Wherry | | PMID:23516357 |
| Chemical compound, drug | Tamoxifen | Sigma-Aldrich | T5648 | |
| Commercial assay or kit | Click-iT Plus EdU Alexa Fluor 488 Flow Cytometry Assay Kit | Thermo Fisher | C10632 | |
| Commercial assay or kit | RNAscope Fluorescent Multiplex Reagent Kit v2 | ACD | 323100 | |
| Commercial assay or kit | PureLink RNA Micro Kit | Thermo Fisher | 12183–016 | |
| Commercial assay or kit | Bioanalyzer High Sensitivity RNA Pico Kit | Agilent | 5067–1513 | |
| Commercial assay or kit | NEBNext Single Cell/Low Input RNA Library Prep Kit for Illumina | New England Biolabs | E6420 | |
| Commercial assay or kit | Bioanalyzer High Sensitivity DNA Kit | Agilent | 5067–4626 | |
| Other | RNAscope probe *mm-Car4-C3* | ACD | 468421-C3 | Probe to detect *Car4* RNA; see *Figure 5—figure supplement 3* and **Materials and methods:** *RNAscope analysis* |

| Reagent type (species) or resource | Designation | Source or reference | Identifiers | Additional information |
|---|---|---|---|---|
| Other | RNAscope probe *mm-Pdgfra-C2* | ACD | 480661-C2 | Probe to detect *Pdgfra* RNA; see *Figure 2—figure supplement 2* and Materials and methods: *RNAscope analysis* |

## Animals

All animal procedures were approved by the Institutional Animal Care and Use Committee of the University of Pennsylvania. Mice were housed in groups of up to five animals per cage when possible and supplied with extra enrichment if singly housed. *ROSA26*[LSL-tdTomato] reporter mice (Gt(ROSA)26Sor[tm14(CAG-tdTomato)Hze]) (*Madisen et al., 2010*) were obtained from the Jackson Laboratory (strain #007914). *Cdh5*[CreERT2] mice (C57BL/6-Tg(Cdh5-cre/ERT2)1Rha) (*Sörensen et al., 2009*) were obtained from Taconic (strain #13073). *Atf3*[lox] mice (Atf3[tm1.1Hai]) (*Wolford et al., 2013*) were a kind gift from Dr. Tsonwin Hai. *Atf3*[CreERT2] mice (Atf3[tm1.1(cre/ERT2)Msra]) (*Denk et al., 2015*) were a kind gift from Dr. Matt Ramer. ARRIVE Essential 10 guidelines were followed for all animal experiments described.

## Tamoxifen delivery

Tamoxifen (Sigma-Aldrich #T5648) was suspended in 1:10 ethanol:corn oil at 20 mg/mL. This solution was administered to mice with either one or two doses of 200 mg/kg (10 μL/g) by oral gavage.

## Influenza injury

The PR8-GP33 H1N1 influenza virus was a gift from Dr. E. John Wherry. Mice were given a titrated dose of 1 $LD_{50}$ (determined empirically in our laboratory) diluted in sterile saline and administered intranasally in a 50 μL dose. In our animals, this dose results in a moderate level of tissue damage with different zones of injury as previously reported (*Liberti et al., 2022*; *Liberti et al., 2021*; *Zacharias et al., 2018*). Control mice were given 50 μL sterile saline, and also administered intranasally. Animals were randomly assigned to PBS or H1N1 groups in an unblinded fashion. Mice were weighed daily for 14 days after infection and euthanized if their weight loss exceeded 30% of their original body weight, calculated as the average of their weight over the first three days following infection. Exclusion criteria were pre-established: mice that lost less than 10% of body weight were not used for experiments. Each experiment was repeated at least two times in the laboratory to ensure consistent results. Tissue was harvested at the time points described for histology or single-cell isolation.

## EdU administration

EdU was dissolved in tap water at 2 g/L and sterile filtered to create a stock solution that was stored at 4 °C. Prior to administration, the stock was diluted 1:10 in tap water to create a solution of 0.2 g/L EdU and sterile-filtered a second time. Mice were allowed to consume EdU water ad libitum over the course of 6–7 days as indicated. EdU was detected using the Click-iT Plus EdU Alexa Fluor 488 Flow Cytometry Assay Kit (Thermo Fisher # C10632) according to the manufacturer's protocol.

## Histology and immunofluorescence

At the indicated time point of tissue analysis, mice were euthanized with a lethal dose of $CO_2$ and cervically dislocated. Lungs were cleared of blood using right ventricle perfusion with PBS. The trachea was cannulated, and the lungs were inflation-fixed with 2% paraformaldehyde at a pressure of 30 cm of water. Fixation continued by immersion in 2% PFA, with shaking, overnight at 4 °C. Tissue was dehydrated through 6 × 30 min washes in 1 X PBS followed by overnight washes in 70% and 95% ethanol and 2 × 1 hr washes in 100% ethanol, all at 4 °C with shaking. Lung tissue was paraffin embedded for microtome sectioning. Hematoxylin and eosin (H&E) staining was utilized to assess tissue structure. Immunohistochemistry was used to recognize various antigens using the antibodies described in the Key Resources table above. Slides were mounted using either Vectashield Antifade Mounting Medium (Vector Laboratories #H-1000) or Slowfade Diamond Antifade Mountant (Thermo Fisher #S36972).

## RNAscope analysis

Lung tissue was prepared as described for histology. RNAscope was performed using the Fluorescent Multiplex Reagent Kit v2 (ACD #323100) according to the manufacturer's instructions. RNAscope probes used are described in the Key Resources table above.

## Imaging and image analysis

Fluorescent images were acquired at 20 X and 60 X using an LSM 710 laser scanning confocal microscope (Zeiss). Cell counting was performed using the 'Cell Counter' macro in FIJI (Plugins→Analyze→Cell Counter). Alveolar area quantification was performed using FIJI as previously described (*Liberti et al., 2021*). Briefly, images captured of IF for PDPN at 20 X were processed using simple thresholding, binarized, subjected to dilation and erosion to remove background, and inverted. Quantification was performed using the Analyze Particles function (Analyze→Analyze Particles) with a lower size limit of 10 $\mu m^2$. Analysis was performed only on tissue regions that were not densely packed with immune cells and did not contain airways or large vessels. Brightfield images of H&E-stained tissue were acquired at 20 X using the EVOS FL Auto 2 Imaging System. Mean linear intercept analysis was performed using MATLAB as previously described (*Liberti et al., 2021*). Briefly, images were sorted manually to avoid large airways and vessels and focus on alveolar regions in the distal lung. At least ten images per mouse were selected for analysis. Quantification was performed using custom MATLAB software described previously (*Paris et al., 2020*; *Obraztsova et al., 2020*).

## Single-cell isolation, flow cytometry, and fluorescence-activated cell sorting (FACS)

Following the dissection and washing protocol outlined above, lung lobes were dissected from the main-stem bronchus and minced with dissecting scissors followed by a razor blade. Tissue was digested in a mixture of collagenase-I (480 U/mL), dispase (100 μL/mL), and DNase (2 μL/mL). Following trituration and filtration, red blood cells were lysed using ACK lysis buffer, and single cells were resuspended in a FACS buffer containing 1% FBS in PBS. Fluorescently conjugated antibodies used for flow cytometry are described in the Key Resources table above. For proliferation analysis, cells that had incorporated EdU were fluorescently labeled using the Click-iT Plus EdU Alexa Fluor 488 Flow Cytometry Assay Kit (Thermo Fisher # C10632) according to the manufacturer's protocol. All flow cytometry analysis was performed using an LSR Fortessa (BD Biosciences) and FACSDiva software. FACS was performed using the FACSJazz (BD Biosciences) and MoFlo Astrios CQ (Beckman Coulter) cell sorting systems.

## Analysis of scRNA-seq data

Previously published scRNA-sequencing data (*Niethamer et al., 2020*) was reanalyzed according to the following parameters. Reads were aligned to the mouse reference genome (mm39/mGRC39) and unique molecular identifier (UMI) counts were obtained using STAR-Solo (v2.7.9a). For further processing, integration, and downstream analysis, we used Seurat v4.0.6. Cells with fewer than 200 genes, greater than 2 median absolute deviations above the median, and with potential stress signals of greater than 5% mitochondrial reads were removed. The cell cycle phase prediction score was calculated using the Seurat function CellCycleScoring. Data were normalized and scaled using the SCTransform function, and the effects of the percent fraction of mitochondria, number of features per cell, and number of UMIs per cell were regressed out. Integration of individual samples was performed using normalized values from SCTransform and the top 3,000 variable genes as anchors for canonical correlation analysis using Seurat v.4. Linear dimension reduction was done via PCA, and the number of PCA dimensions was evaluated and selected based on the assessment of an ElbowPlot. Data were clustered using the Louvain graph-based algorithm in R and cluster resolution was chosen based on evaluation by the clustree program (*Kiselev et al., 2017*). The Uniform Manifold Projection (UMAP) data reduction algorithm was used to project the cells onto two-dimensional coordinates. Clusters were then assigned cell types based on annotation with canonical marker genes. For intra-cluster gene expression differences, the FindMarkers function was used to identify variation between specified clusters, and the resultant gene sets were comparted via the MAST method. GO enrichment analysis was done with the clusterProfiler R package.

## RNA isolation and RNA sequencing

After FACS to isolate tdTomato⁺ ECs, cells were pelleted and resuspended in RNA lysis buffer before isolation of total RNA using the PureLink RNA Micro Kit (Thermo Fisher #12183–016) according to the manufacturer's instructions. RNA concentration and integrity were determined using the Bioanalyzer High Sensitivity RNA Pico Kit (Agilent #5067–1513). Sequencing libraries were prepared using the NEBNext Single Cell/Low Input RNA Library Prep Kit for Illumina (New England Biolabs #E6420) according to the manufacturer's instructions. cDNA and library quality were assessed using the Bioanalyzer High Sensitivity DNA Kit (Agilent #5067–4626). Libraries were sequenced using the Illumina HiSeq. Fastq files were aligned to the mouse reference genome (mm39/mGRC39) using the STAR aligner v2.7.9a. Duplicate reads were removed using MarkDuplicates from Picard tools, and per gene read counts for Ensembl (v104) gene annotations were computed. Expression levels in counts per million (CPM) were normalized and transformed using VOOM in the limma R package. To account for sources of latent variation such as batch effects, surrogate variables were calculated using the svaseq function from the R SVA package. Differential gene expression analysis was conducted using the limma package. Gene Ontology and pathway analysis were performed using the clusterProfiler v4.4.4. All plots were constructed in R using ggplot2 or Complexheatmap. *qRT-PCR.* RNA was isolated as described above, and cDNA was obtained using SuperScript IV VILO Master Mix (Thermo Fisher) according to the manufacturer's instructions. qRT-PCR was performed using Power SYBR Green 2 x Master Mix (Thermo Fisher) on a QuantStudio 7 Flex system (Applied Biosystems). Data were analyzed using a standard curve method, and all calculations incorporated primer efficiencies calculated from standard curves and were normalized to the input number of cells. Primer sequences:

> *Atf3* Forward, 5'- TGGAGATGTCAGTCACCAAGT-3';
> *Atf3* Reverse, 5'-TTCTTCAGCTCCTCAATCTGGG-3';
> *Fgfr1* Forward, 5'- AGCGCCAAGTGAGAGTCAG-3';
> *Fgfr1* Reverse, 5'- CTCCACTTCCACAGGGACTC-3';
> *Lrp5* Forward, 5'- TTCCAACATGCTGGGTCAGG-3';
> *Lrp5* Reverse, 5'- GCCCGTTCAATGCTATGCAG-3';
> *Wif1* Forward, 5'- CCCGATGTATGAACGGTGGT-3';
> *Wif1* Reverse, 5'- GGTGGTTGAGCAGTTTGCTTT-3'.

## Statistical analysis

Statistical analysis of scRNA-seq and RNA-seq data was performed in R. All other statistical analysis was performed using GraphPad Prism software. Unpaired t-tests (parametric) and Mann-Whitney rank-sum tests (nonparametric) were used to assess differences between the two groups. If more than two groups were analyzed, a Kruskal-Wallace test was used to determine if differences between groups were statistically significant, followed by Dunn's multiple comparison tests to compare individual groups. Graphs are displayed as bar plots indicating mean and standard deviation. For data quantification purposes, each dot on a graph represents a single mouse, i.e., a biological replicate. p-values are indicated in figure legends.

## Acknowledgements

The authors would like to acknowledge the core facilities at Penn without whom we could not have completed this work, including the Penn Cytomics and Cell Sorting Resource Laboratory, the CHOP Flow Cytometry Core, and the Penn Cell and Developmental Biology Microscopy Core. We thank Dr. Tsonwin Hai for the Atf3ˡᵒˣ mouse line and Dr. Matt Ramer for the Atf3^CreERT2 mouse line. We would also like to acknowledge Dr. E John Wherry for providing the PR8 H1N1 influenza strain. We also thank the members of the Morrisey lab for their helpful suggestions and discussions over the course of this work. The Morrisey lab is supported by grant funding from the National Institutes of Health and the Longfonds Foundation of the Netherlands.

# Additional information

## Competing interests

Su Zhou, Edward E Morrisey: Reviewing editor, eLife. The other authors declare that no competing interests exist.

## Funding

| Funder | Grant reference number | Author |
| --- | --- | --- |
| National Heart, Lung, and Blood Institute | F32HL152664 | Terren K Niethamer |
| National Heart, Lung, and Blood Institute | K99HL164960 | Terren K Niethamer |
| National Heart, Lung, and Blood Institute | R01HL152194 | Edward E Morrisey |
| National Heart, Lung, and Blood Institute | R01HL132999 | Edward E Morrisey |
| National Heart, Lung, and Blood Institute | U01HL134745 | Edward E Morrisey |
| National Heart, Lung, and Blood Institute | U01HL148857 | Edward E Morrisey |
| Longfonds | BREATH Consortium | Edward E Morrisey |

The funders had no role in study design, data collection and interpretation, or the decision to submit the work for publication.

## Author contributions

Terren K Niethamer, Conceptualization, Formal analysis, Validation, Investigation, Visualization, Writing – original draft, Writing – review and editing; Lillian I Levin, Investigation, Writing – original draft; Michael P Morley, Resources, Data curation, Software, Formal analysis, Validation, Visualization, Methodology, Writing – review and editing; Apoorva Babu, Resources, Data curation, Software, Formal analysis, Visualization, Methodology; Su Zhou, Resources, Investigation; Edward E Morrisey, Conceptualization, Resources, Supervision, Funding acquisition, Project administration, Writing – review and editing

## Author ORCIDs

Terren K Niethamer http://orcid.org/0000-0002-0914-994X
Edward E Morrisey http://orcid.org/0000-0001-5785-1939

## Ethics

All animal experiments were approved by the Institutional Animal Care and Use Committee (IACUC) of the University of Pennsylvania under the protocol #806345. The study was performed according to the recommendations laid out in the Guide for the Care and Use of Laboratory Animals from the National Institutes of Health. Every effort was made to minimize animal suffering; specifically, influenza administration was performed under ketamine/xylazine anesthesia, and animals with 30% body weight loss or greater after influenza infection were euthanized.

## Decision letter and Author response

Decision letter https://doi.org/10.7554/eLife.83835.sa1
Author response https://doi.org/10.7554/eLife.83835.sa2

# Additional files

## Supplementary files

• MDAR checklist

## Data availability

The scRNA-sequencing data described here is previously published (*Niethamer et al., 2020*) and is deposited in the NCBI GEO Repository under accession number GSE128944. RNA sequencing data of ATF3 wild-type and knockout endothelial cells is deposited under accession number GSE213475.

The following dataset was generated:

| Author(s) | Year | Dataset title | Dataset URL | Database and Identifier |
|---|---|---|---|---|
| Niethamer TK, Levin LI, Morley MP, Babu A, Zhou S, Morrisey EE | 2022 | ATF3 promotes endothelial cell response to acute lung injury | https://www.ncbi.nlm.nih.gov/geo/query/acc.cgi?acc=GSE213475 | NCBI Gene Expression Omnibus, GSE213475 |

The following previously published dataset was used:

| Author(s) | Year | Dataset title | Dataset URL | Database and Identifier |
|---|---|---|---|---|
| Niethamer TK, Stabler CT, Morley MP, Morrisey EE | 2020 | Mapping pulmonary endothelial cell heterogeneity at homeostasis and during tissue regeneration | https://www.ncbi.nlm.nih.gov/geo/query/acc.cgi?acc=GSE128944 | NCBI Gene Expression Omnibus, GSE128944 |

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
