## [Editor Report]

The reviewers concurred that your study has advanced our mechanistic understanding of lung regeneration. While the importance of regeneration of alveolar capillaries for long response to injury has been long recognized, the regulation of this process has not been well understood. You provide novel, comprehensive, and compelling evidence that the expression of the transcription factor Atf3 in alveolar capillary endothelial cells plays a critical role in the regeneration of alveolar capillaries following lung injury.

---

## [Decision Letter]

**Decision letter after peer review:**

Thank you for submitting your article "Atf3 defines a population of pulmonary endothelial cells essential for lung regeneration" for consideration by *eLife*. Your article has been reviewed by 3 peer reviewers, and the evaluation has been overseen by a Reviewing Editor and Paul Noble as the Senior Editor. The reviewers have opted to remain anonymous.

Essential revisions:

1. Please define the extent of H1N1-mediated inflammation in Atf3-deficient mice, and clarify why the aberrant induction of metaplastic Krt5+"pods" by influenza infection is not seen in these mice and how preventing expansion of the CAP1B population would not reveal this feature of the H1N1 lung phenotype.

2. Please consider integrating the scRNA-seq data analysis and interpretation that have been performed separately for the control lung and H1N1 infected lung, reanalyzing the data, and determining the differential gene expression between these two groups.

3. Please verify and quantify the deletion of Atf3 in the mutant mouse.

4. Please clarify how exactly were the regions of viral damage selected

5. In Figure 1S2, Atf3+ CAP1_B is mostly negative for proliferative gene Mki67. The discrepancy between this data and the EdU results needs to be explained. Also, in Figure 1A&B, Atf3 is expressed by the population of venous ECs that seems to increase after injury. Please clarify if these cells may contribute to the increased CD31+/EdU+ population.

*Reviewer #1 (Recommendations for the authors):*

I have only a few comments for clarification and improvement of the manuscript.

1. The authors point to several reports implicating Atf3 in controlling the inflammatory response post-injury and they find an increase in cytokine expression in Atf3 null lungs post-influenza. However, it is not clear from the lung sections (H&E and IF) the extent to which H1N1-mediated inflammation is present in Atf3-deficient mice.

2. Please clarify why the aberrant induction of metaplastic Krt5+"pods" by influenza infection is not seen in Atf3-deficient mice (which show abnormal alveolar repair). It is unclear how preventing the expansion of the CAP1B population would not reveal this feature of the H1N1 lung phenotype.

3. Although ATf3 is enriched in CAP1B ECs, the U-Map representation of Atf3 (Figure 1.S1B) shows still a sizable population of CAP1A cells expressing this transcription factor (also: dot plot in S1H ~60% compared to ~80% of CAP1B cells). Moreover, Atf3 is also increased in the venous EC population of H1N1-infected lungs. Can the authors better define the contribution of the non-CAP1B cells to the alveolar regeneration described here given that the Atf3 reporter line will label collectively these cells?

4. In Figure 2F-G: please provide a higher resolution picture for both PBS and H1N1 panels including EndoM1 staining shown in a separate channel to illustrate more clearly the expansion of the ATF3 lineage labeled ECs.

5. iI Figure 3D, G the authors use alveolar area (umm2) as a parameter to assess alveolar destruction or inability to repair, however in separate experiments (Figure 3, S2E) Mean Linear Intercept was used, instead. Was there any specific reason for that, since both are intended to detect integrity/ enlargement of alveolar spaces?

*Reviewer #2 (Recommendations for the authors):*

Main concerns:

– The authors should reanalyze their scRNA-seq data after integration of both datasets, and clarify their cluster annotations, specifically how Cap1_A and Cap1_B were identified. Differential gene expression between these two populations would be informative for the readers.

– It is of extreme importance that the authors verify the deletion of Atf3 in their mutant mouse and quantify it.

– It would benefit the field and strengthen the impact of the manuscript if the authors assessed the alveolar changes in their model considering other cell types, specifically endothelial cells.

– It is crucial that more clarification is provided as to how the regions of viral damage were selected.

*Reviewer #3 (Recommendations for the authors):*

There are several points that need to be considered:

1. Based on feature plots in Figure 1S2, Atf3+ CAP1_B is mostly negative for proliferative gene Mki67. The discrepancy between this data and the EdU results needs to be explained.

2. In Figures 1A and B, Atf3 is also expressed by venous ECs, and the venous population seems to increase after injury. These cells may contribute to the increased CD31+/EdU+ population as well.

3. Some validation of signaling or downstream genes in mouse lungs is required in Figure 4.

4. The underlying mechanism of Atf3-medicated apoptosis and proliferation inhibition in lung ECs is unclear.

5. In Figure 1, control and H1N1-infected lung datasets were analyzed separately. The annotation may need to be done after the integration of the two datasets first to keep things consistent.

6. Flow cytometry plots are lacking in Figure 2C-E.

---

## [Author Response]

Essential revisions:1. Please define the extent of H1N1-mediated inflammation in Atf3-deficient mice, and clarify why the aberrant induction of metaplastic Krt5+"pods" by influenza infection is not seen in these mice and how preventing expansion of the CAP1B population would not reveal this feature of the H1N1 lung phenotype.

We agree that it is essential both to define the extent of H1N1-mediated inflammation in Atf3 wild-type and knockout mice and to compare this factor between genotypes. We have therefore used a previously published method for quantifying regions of severe, damaged, and normal tissue structure (Liberti et al., *Cell Reports* 2021) in both Atf3 wild-type and knockout animals. Our results show that Atf3 wild-type and knockout mice have similar levels of tissue damage, and we have added a supplemental figure demonstrating these data (new Figure 3 —figure supplement 2). We have also clarified how regions were selected for quantification of alveolar area (please see point #4 below).

2. Please consider integrating the scRNA-seq data analysis and interpretation that have been performed separately for the control lung and H1N1 infected lung, reanalyzing the data, and determining the differential gene expression between these two groups.

We have integrated the control and H1N1-infected scRNA-seq datasets and reanalyzed the integrated data. We then analyzed CAP1_A and CAP1_B populations, comparing their gene expression between control and influenza conditions. Unbiased clustering of the integrated dataset reveals the same clusters we identified in the individual datasets, with cells from control and flu contributing to each cluster (with the exception of proliferating endothelial cells, which are found only in the H1N1-infected lung). We have added a supplemental figure outlining these data (Figure 1 —figure supplement 3).

3. Please verify and quantify the deletion of Atf3 in the mutant mouse.

We agree that this is an important quantification to make. We have performed qRT-PCR for *Atf3* in both the animals used to perform the RNA sequencing experiment as well as a new cohort of animals to confirm *Atf3* deletion. We have added these results to a new supplemental figure accompanying Figure 4 (Figure 4 —figure supplement 1).

4. Please clarify how exactly were the regions of viral damage selected

H1N1 influenza injury in mice is heterogeneous, with regions of severe alveolar destruction marked by densely packed immune cells, adjacent regions of damaged tissue, and regions of tissue that appear to have normal tissue structure, as we and others have previously described (Zacharias, Frank et al., *Nature* 2018; Liberti et al., *Cell Reports* 2021; Niethamer et al., *eLife* 2020). However, it has become increasingly apparent that these regions where tissue structure appears normal are actually regions of active regeneration, and endothelial cell proliferation is increased in these regions (Niethamer et al., *eLife* 2020). We therefore selected 20X fields in these areas to use for quantifying alveolar area, as these are actively regenerating regions where alveolar structures are present for quantification. Because of the changes to tissue structure seen in damaged or destroyed tissue areas, we did not select these regions for quantification, although they were present at similar frequency in Atf3 wild-type and knockout animals (point #1 above).

5. In Figure 1S2, Atf3+ CAP1_B is mostly negative for proliferative gene Mki67. The discrepancy between this data and the EdU results needs to be explained. Also, in Figure 1A&B, Atf3 is expressed by the population of venous ECs that seems to increase after injury. Please clarify if these cells may contribute to the increased CD31+/EdU+ population.

The difference between the expression of the gene *Mki67* in our scRNA-seq data and the EdU incorporation in vivo can be explained by the differences in methodology. Our scRNA-seq data is collected at a single time point of 14 days post infection (dpi), whereas animals treated with EdU received it in their drinking water over the course of one week from 7-14 dpi or from 14-21 dpi. These time windows were chosen based on the increases in endothelial proliferation seen during these time periods in Niethamer et al., *eLife* 2020. In addition, the proliferating endothelial cells cluster separately in the scRNA-seq analysis and also express *Atf3*. To further clarify the relative expression of *Atf3* in different endothelial subtypes, we have also added violin plots to Figure 1 demonstrating relative *Atf3* expression in each endothelial cell cluster.

Reviewer #1 (Recommendations for the authors):I have only a few comments for clarification and improvement of the manuscript.1. The authors point to several reports implicating Atf3 in controlling the inflammatory response post-injury and they find an increase in cytokine expression in Atf3 null lungs post-influenza. However, it is not clear from the lung sections (H&E and IF) the extent to which H1N1-mediated inflammation is present in Atf3-deficient mice.2. Please clarify why the aberrant induction of metaplastic Krt5+"pods" by influenza infection is not seen in Atf3-deficient mice (which show abnormal alveolar repair). It is unclear how preventing the expansion of the CAP1B population would not reveal this feature of the H1N1 lung phenotype.

Please see our response to the point #1 above.

3. Although ATf3 is enriched in CAP1B ECs, the U-Map representation of Atf3 (Figure 1.S1B) shows still a sizable population of CAP1A cells expressing this transcription factor (also: dot plot in S1H ~60% compared to ~80% of CAP1B cells). Moreover, Atf3 is also increased in the venous EC population of H1N1-infected lungs. Can the authors better define the contribution of the non-CAP1B cells to the alveolar regeneration described here given that the Atf3 reporter line will label collectively these cells?4. In Figure 2F-G: please provide a higher resolution picture for both PBS and H1N1 panels including EndoM1 staining shown in a separate channel to illustrate more clearly the expansion of the ATF3 lineage labeled ECs.

We have added violin plots to Figure 1, which we feel will better represent the greater *Atf3* expression in CAP1_Bs relative to other endothelial cell subtypes. The reviewer is correct that Atf3-expressing cells are found in large vessels, but they are also numerous in the alveolar capillary space and increase with influenza in these regions. We have added lower-magnification, higher-resolution images of *Atf3^CreER^; ROSA26^tdTomato^* animals, both control and influenza-infected, to illustrate this expansion in a new Figure 2 —figure supplement 3. This increase is also quantified in Figure 2C. We have also clarified this in the text.

5. iI Figure 3D, G the authors use alveolar area (umm2) as a parameter to assess alveolar destruction or inability to repair, however in separate experiments (Figure 3, S2E) Mean Linear Intercept was used, instead. Was there any specific reason for that, since both are intended to detect integrity/ enlargement of alveolar spaces?

The mean linear intercept algorithm used in Figure 3 —figure supplement 2 does not perform well on influenza-infected tissue because of the heterogeneity in tissue structure seen in the H1N1-infected animals. We therefore used it for the homeostasis experiment shown in this figure. For the sake of consistency, we have repeated our analysis of these animals using alveolar area quantification and have added this analysis to the existing figure. We have added to the text to clarify this point and have also clarified our choice of regions for both types of analysis (please see point #4 above).

Reviewer #3 (Recommendations for the authors):There are several points that need to be considered:1. Based on feature plots in Figure 1S2, Atf3+ CAP1_B is mostly negative for proliferative gene Mki67. The discrepancy between this data and the EdU results needs to be explained.

Please see our response to point #5 above.

2. In Figures 1A and B, Atf3 is also expressed by venous ECs, and the venous population seems to increase after injury. These cells may contribute to the increased CD31+/EdU+ population as well.

Please see our response to point #5 above.

3. Some validation of signaling or downstream genes in mouse lungs is required in Figure 4.

We have added qRT-PCR analysis of several downstream genes to a new Figure 4 —figure supplement 1.

4. The underlying mechanism of Atf3-medicated apoptosis and proliferation inhibition in lung ECs is unclear.

We have added to the text of the Discussion section to further discuss this point.

5. In Figure 1, control and H1N1-infected lung datasets were analyzed separately. The annotation may need to be done after the integration of the two datasets first to keep things consistent.

Please see our response to point #2 above.

6. Flow cytometry plots are lacking in Figure 2C-E.

We have added representative plots to Figure 2 —figure supplement 1.